# The First K^+^-Channel Blocker Described from *Tityus fasciolatus* Venom: The Purification, Molecular Cloning, and Functional Characterization of α-KTx4.9 (Tf5)

**DOI:** 10.3390/toxins17020096

**Published:** 2025-02-18

**Authors:** Isolda de Sousa Monteiro, Israel Flor Silva de Araújo, Thalita Soares Camargos, Ernesto Ortiz, Adolfo Carlos Barros de Souza, Jonathan Dias Lima, Lourival D. Possani, Elisabeth Ferroni Schwartz, Diogo Vieira Tibery

**Affiliations:** 1Laboratory of Neuropharmacology, Institute of Biological Sciences, University of Brasilia, Brasilia 70910-900, Brazil; isoldamonteiro31@gmail.com (I.d.S.M.); israelfsaraujo@gmail.com (I.F.S.d.A.); thalitasoares@gmail.com (T.S.C.); adolfo_quimica@hotmail.com (A.C.B.d.S.); dias12lima@gmail.com (J.D.L.); efschwa@unb.br (E.F.S.); 2Colégio Militar de Brasília, Brasília 70790-020, Brazil; 3Institute of Biotechnology, National Autonomous University of Mexico, Cuernavaca 62210, Mexico; ernesto.ortiz@ibt.unam.mx (E.O.); lourival.possani@ibt.unam.mx (L.D.P.)

**Keywords:** KTxs, K^+^-channel, scorpion, Kv1.2, Kv1.3, Tf5, *Tityus fasciolatus*, α-KTx4.9

## Abstract

Hundreds of toxins, particularly from scorpions of lesser medical significance, remain unknown, especially those from species endemic to specific ecosystems, such as *Tityus fasciolatus*. Their discovery could contribute to the development of new drugs for channelopathies and other diseases. Tf5 is a new peptide that has been identified from the venom of *Tityus fasciolatus*, a scorpion species endemic to the Brazilian Cerrado ecosystem. A full-length cDNA sequence of the Tf5 gene was obtained through a previously constructed transcriptomic library, where an ORF (Open Reading Frame) sequence with a length of 180 was found, including the 37 aa mature KTx domain, which has six Cys residues. Tf5 was purified from the crude venom, resulting in a peptide with a molecular mass of 3983.95 Da. Its K^+^ channel blocker activity was evaluated on Kv1.1, Kv1.2, Kv1.3, and Kv1.4 subtypes. Of these Kv channels, the peptide demonstrated an ability to block Kv1.2 and Kv1.3 with an IC_50_ of 15.53 nM and 116.41 nM, respectively. Additionally, Tf5 shares a high degree of sequence identity with toxins from the α-KTx4 subfamily, which led to it being classified as α-KTx4.9. This is the first Kv channel blocker described from the *T. fasciolatus* scorpion.

## 1. Introduction

With estimates that they evolved over 400 million years ago, scorpions constitute a remarkable ancient group of terrestrial organisms [1]. Adaptable to nearly all habitats, they are present on all continents, except Antarctica [2]. Among the families of scorpions that exist in Brazil, the Buthidae family stands out [3]. It comprises 82 species distributed across 8 genera, with the genus *Tityus* being of the utmost clinical importance, encompassing various representatives such as *Tityus serrulatus, T. bahienses, T. stigmurus, T. obscurus, T. trivittatus, and T. fasciolatus* [4,5]. Due to the large number of scorpionism cases they are involved in, they receive more attention from researchers [3,6]. The venom of these scorpions is composed of various molecules, including peptides, enzymes, inorganic salts, amines, nucleotides, and others [7,8]. Scorpion peptides present different activities, such as antimicrobial, antifungal, antiprotozoal, antiviral, immunomodulatory, anticancer, ion channel-modulating activities, and others, due to an assortment of compounds whose activities remain unknown [8,9].

*Tityus fasciolatus* is found in the Cerrado biome, inhabiting areas with low human interference, where it is frequently associated with *Armitermes euamignathus* termites [10]. *T. fasciolatus* venom contains a wide variety of peptides that are responsible for the typical symptoms seen in cases of envenoming, which present with a much lower severity compared to other species within the same genus [11,12,13], such as *T. serrulatus*, which is the main species responsible for scorpion accidents in Brazil. From the venom of *T. fasciolatus* five toxins capable of modulating sodium channels have been described: Tf1, Tf3, Tf4a [11], Tf2 [14], and Tf1a [15]. As far as we know, toxins with the ability to act on potassium channels have yet to be characterized. According to a comprehensive transcriptomic and proteomic study (unpublished), 48% of the venom from this species is composed of toxins that could potentially act on K^+^ channels, suggesting the existence of a considerable range of peptides yet to be described in the venom of *Tityus fasciolatus* [14].

Scorpion toxins acting on voltage-gated potassium ion channels (KTx) are organized into seven families, α, β, γ, δ, ε, λ, and k-KTx [16,17], distributed according to their homology, three-dimensional folding pattern, and activity. Those with the classical cysteine-stabilized α/β (CSα/β) folding are grouped into three main families—α-KTx (consisting of approximately 20–45 residues and containing 3–4 disulfide bridges) [18,19], β-KTx (45–80 residues and 3 disulfide bridges) [18,20], and γ-KTx (35–45 residues and 3–4 disulfide bridges)—that affect a specific subset of Kv channels known as ERG channels [18,21]. Toxins with the cysteine-stabilized helix–loop–helix folding (CSα/α) are classified as being within the κ-KTx family (25–30 residues and 2 disulfide bridges) [22,23]. Kunitz toxins are known as δ-KTx (60–70 residues and 3–4 disulfide bridges) [24]. Finally, toxins sharing an inhibitory cystine knot (ICK) motif comprise the λ-KTx (35–40 residues and 3 disulfide bridges) [25] and ε-KTx families (29 residues and 4 disulfide bridges) [26].

Voltage-gated potassium channels (Kv) are involved in the release of neurotransmitters and hormones [27], smooth muscle contractility [28], cardiac muscle excitability [29], and other important physiological functions [30,31]. Kv channels are tetrameric channels formed by four alpha subunits; each one of the subunits is composed of six transmembrane segments [32]. These alpha subunits are formed of either identical (homotetramers) or different subunits (heterotetramers), and they form a channel structure that consists of two regions: the pore region, formed by segments S5 and S6, which are connected by the pore loop (P), and the voltage sensor region, formed by four transmembrane segments S1–S4 [33,34]. Beta subunits influence channel kinetics when co-expressed [35].

Disorders related to the malfunction of ion channels are called channelopathies, and these can result from mutations in the genes encoding ion channels, alterations in ion channel synthesis, and post-translational modifications [36]. Examples of Kv-channel channelopaties include episodic ataxia type 1, acquired neuromyotonia [37], Long QT Syndrome [38], and Jervell and Lange-Nielsen syndrome, among others [39].

Kv1.2 (KCNA2), a potassium voltage-gated channel, is significantly expressed in the central and peripheral nervous systems [40]. Mutations in KCNA2 can lead to gain-of-function, loss-of-function, or mixed variants, causing disorders such as epilepsy, intellectual disability, ADHD, autism, pain syndromes, autoimmune diseases, and movement disorders [40]. Kv1.1 and Kv1.2 channels are found in the spinal dorsal horn and dorsal root ganglia [41], where they regulate sensory neuron activity during pain signaling by reducing excitability through K^+^ efflux. In mammals, Kv1.1 and Kv1.2 channels, together with Kv1.5 and Kv3.1, also play key roles in sperm motility, volume regulation, and capacitation by controlling K^+^ efflux and membrane hyperpolarization [42].

Kv1.3 channels are present in different types of cells throughout the body and expressed in the brain, lymph nodes, spinal cord, kidneys, and lungs, with large numbers also seen in human lymphocytes [43]. Due to their presence in diverse cell types, these channels are involved in several physiological processes, such as cytokine production and the modulation of calcium signaling to induce T cell proliferation, playing important roles in many autoimmune diseases [44,45,46,47,48].

This study characterizes the cDNA-derived precursor sequence of a novel potassium channel blocker from the venom of *Tityus fasciolatus* and establishes its classification among scorpion toxins. These findings enhance our current understanding of potassium channel-targeting peptides, which remain largely unexplored in this species. Given the critical physiological role of Kv channels in various biological processes and pathological conditions, this research provides insights into ion channel modulators.

## 2. Results

### 2.1. Venom Gland Transcriptome

A transcriptomic analysis performed after the construction of the cDNA library from *T. fasciolatus* venom glands allowed for the identification of the complete coding sequence of the Tf5 precursor. The 180 nucleotide-long CDS encodes a peptide of 59 amino acid residues (Figure 1), including a signal peptide of 22 amino acids (MKAFYGILIIFIFISMLDLSQQ) and a mature segment with 37 residues (VFINARCRGSPECLPKCKEAIGKSAGKCMNGKCKCYP), whose monoisotopic theoretical molecular mass is [M + H]^+^ = 3983.88 Da.

### 2.2. Toxin Purification

Samples containing 3.5 mg of crude venom from the scorpion *Tityus fasciolatus* were fractionated by RP-HLPC as previously described [14]. The chromatographic fraction that eluted at 33.5% acetonitrile during crude venom fractionation (Figure 2A) went through an additional purification step that employed RP-HPLC until a pure peptide named Tf5 was obtained (Figure 2B).

### 2.3. Molecular Mass Determination and MS/MS Sequencing

The partial sequence was obtained by the In-source Decay method (ISD) using MALDI TOF/TOF equipment, and the spectrum generated (Appendix A) by this fragmentation resulted in a sequence that coincided with 23 amino acids (SPECLPKCKEAIGKSAGKCMNGK) present in the middle region of the mature Tf5 peptide previously deduced by transcriptomic analysis.

The purified Tf5 had a monoisotopic mass of [M + H]^+^ = 3983.95 Da, [M + 2H]^2+^ = 1993.57 Da (Figure 3), which was close to the theoretical monoisotopic molecular mass [M + H]^+^ calculated for the mature Tf5 predicted from a cDNA-translated precursor ([M + H]^+^ = 3983.88 Da). Only peaks corresponding to the different charges of Tf5 were detected in the mass spectrometry, demonstrating the high purity of the peptide.

### 2.4. Alignment and Structure

The search for other peptides with sequence similarity to Tf5 using BLASTP and their subsequent alignment with Clustal Omega showed that Tf5 possesses the conserved domains of the Toxin-2 family (pfam00451), whose members display potassium channel blocking activity, and that it belongs to the α-KTx4 subfamily of scorpion K^+^-channel blockers (Figure 4A). Tf5 showed a 94.59% similarity to the peptides α-KTx4.1 and α-KTx4.7 (which exhibit a 100% sequence identity with each other) described from other Brazilian *Tityus* species. Considering this, Tf5 was classified as α-KTX4.9, the newest member of the α-KTx4 family.

KTx4 are cystine-dense peptides (with a CDP-defining motif [49]) with Cys1-Cys4, Cys2-Cys5, and Cys3-Cys6 disulfide connectivity, and their Cys2-Cys5 disulfide linkage passes through the macrocycle formed by the Cys1-Cys4 and Cys3-Cys6 disulfide linkages. The tertiary structure of the Tf5 peptide was determined using AlphaFold3, a structural prediction model based on machine learning. To validate the predicted models, the structure of the alpha-KTx4.5 toxin (PDB: 6ATN.1) was used as a reference, as it shares 94.59% identity with Tf5. The structure of alpha-KTx4.5, previously determined by X-ray crystallography, revealed a βαββ pattern stabilized by three disulfide bridges (Figure 4B).

In addition to the toxins with an identity within the α-KTx family 4, new putative toxins have been identified in recent years through transcriptome analysis and may possibly be classified as part of the α-KTx family 4 in the future. These include the toxins Ts22, Ts23, Ts24, and Ts25 from *Tityus serrulatus* [50]; Tme5, Tme6, Tme7, and Tme8 from *Tityus melici* [51]; and Tcis20, Tcis21, and Tcis22 from *Tityus cisandinus* [52].

### 2.5. Bioinformatic Analysis to Predict Hemolytic Effects

To investigate certain characteristics of the Tf5 toxin, in silico tests based on its amino acid sequence were performed to analyze its properties and assess its cytotoxicity. The in silico cytotoxicity evaluation completed by a machine learning-based tool, using a human erythrocyte model, predicted a value of 0.63, classifying it as non-hemolytic [53]; a second in silico assay using HLPpred-Fuse software predicted a low probability of the peptide being toxic (0.147) [54].

### 2.6. Electrophysiological Characterization

Considering the spectrum of activity of KTx4 peptides [49,55,56,57,58], the activity of Tf5 was verified, at 500 nM, on voltage-dependent potassium channels Kv1.1 to Kv1.4 by inhibiting the amplitude of a current stimulated at 30 mV (Figure 5). Kv1.2 and Kv1.3 were the only channels tested that showed sensitivity to Tf5 at the test concentration of 500 nM, with inhibitions of 90.23 ± 3.78% (n = 9) in Kv1.2 and 81.80 ± 4.29% (n = 7) in Kv1.3 (Figure 5B,C). Due to its high inhibition rate at a concentration of 500 nM, experiments to establish a dose–response curve were conducted, and IC_50_ of 15.53 ± 1.12 nM for Kv1.2 and 116.41 ± 1.15 nM for Kv1.3 were observed, demonstrating that the Kv1.2 channel is more sensitive to this toxin (Figure 5E,F).

## 3. Discussion

The sequences coding for the toxins obtained from the transcriptomic analysis were grouped and analyzed separately. A total of 278 contigs were found that code for toxins, 89% of which are toxins with probable action on Na^+^ and K^+^ channels. Among these, 12 sequences coded for the Tf5 precursor, defined by a CDS of 177 nucleotides encoding a propeptide of 59 amino acids, with 22 representing the signal peptide, and the remaining 37 residues comprising the predicted mature toxin, including six Cys, which stabilize the structure with three disulfide bridges.

This work reports the nucleotide sequence code of the precursor, as well as the biochemical and physiological characterization of Tf5, the first α-KTx described from *Tityus fasciolatus* venom. Like many peptides from scorpion venoms that modulate ion channels, the primary structure of Tf5 has a high content (22%) of basic amino acids, mainly Lys, which confer a basic character to the peptide (with a theoretical isoelectric point of 9.27).

Based on the resulting alignment of both the precursor and mature sequences with similar peptides identified in scorpions and the criteria proposed by Tytgat et al., 1999 [18], Tf5 was classified as a new K^+^-channel blocker toxin from family 4 and was named KTx4.9. At present, KTx4 peptides have been described only in the venom of South American Buthidae scorpion species, including *Tityus serrulatus* [59,60], *T. stigmurus* [56,61], *T. costatus* [62], *T. obscurus* [58], and *T. fasciolatus* (present work) from Brazil; *T. discrepans* [63] from Venezuela; and *Centruroides margaritatus* [55] from Colombia.

Its high sequence identity with peptides belonging to the α-KTX4 subfamily that have solved crystal or NMR structures [49] suggests that Tf5 adopts a CSα/β structural motif, which is indeed a βαββ three-disulfide-containing scaffold, as implied in the structural modeling of Tf5. Together with other scorpion peptides acting on ion channels, the α-KTX4 subfamily is referred to as “cystine-dense peptides” (CDPs), meaning they are short peptides with independent folding domains, at least three cystines, and a compact structure. The CDP motif is defined as having (1) six or more cysteine residues in a 13–81 amino acid-long molecule, excluding those recognizable either as cytoplasmatic, zinc finger, or growth factor cystine knot proteins, with (2) the distribution of the cysteines constricted to the arrangement Cys-X [0-15]-Cys-X [0-15]-Cys-X [0-15]-Cys-X [0-15]-Cys-X [0-15]-Cys (with a“X” meaning any amino acid) [49]. The consensus sequence for KTx4 peptides has the following Cys distribution: Cys-X [5]-Cys-X [3]-Cys-X [10]-Cys-X [4]-Cys-X [1]. The pairs of cysteine residues in α-KTX4.1 [64] and α-KTX4.5 [49] are Cys1–Cys4, Cys3–Cys6, with the Cys2–Cys5 disulfide bridge passing through the macrocycle formed by the other two cystines. Expecting a similar cystine connectivity as in α-KTX4.1 and α-KTX4.5, the Tf5 disulfide bonds should be formed between Cys7–Cys28, Cys13–Cys33, and Cys17–Cys35. Following the nomenclature for knotted CDPs [49], the Cys-pairing arrangement of KTx4 representatives is known as a hitching [49], after the kind of knots used to tie a rope around an object.

α-Ktx peptides are short polypeptides consisting of 23 to 42 amino acids. They show the highest degree of diversity among the KTx toxins, comprising approximately 180 toxins distributed into 32 toxin subfamilies. In addition to their conserved amino acid sequence and their peptide’s three-dimensional arrangement, multiple characteristics of the primary structure of these toxins are decisive for their selectivity towards K^+^ channels, particularly those belonging to the Shaker family. Scorpion toxins acting on voltage-dependent potassium channels exhibit patterns of conserved regions, including amino acid residues in key positions responsible for their interaction with these ion channels [65]. These toxins have been characterized by the presence of a functional dyad, a pair of amino acids that are the determinants of their interaction with voltage-dependent K^+^ channels of family 1.

Their functional dyad is defined by a Lys and an aromatic or hydrophobic residue, a Tyr, Phe, or Leu located nine positions downstream, and these are separated from each other by 6.6 Å [66,67]. The Lys projects into the Kv selectivity filter and competes with K^+^ ions at a binding site on the extracellular side of the conduction pathway [68], making it critical for blocking the channel with high affinity, while the aromatic residue (mostly Tyr or Phe) seems to determine channel selectivity among the Shaker Kv1 representatives [68]. This functional dyad was originally proposed to be required for the high-affinity blocking of Kv channels in general, yet, with the progress in the data available, it seems to be crucial for the blockage of Kv1.2 with high affinity and does not seem to be that important for the Kv1.3 channels [69].

Determinants of selectivity between Kv1.2 and Kv1.3 in the amino acid sequence of α-KTx4 have been proposed [56,70]. Increased activity toward Kv1.3 is suggested to be due to the overall positive charge of a toxin, basic residues in its C-terminal region, non-tyrosine as the aromatic residue of its dyad, and a Met residue two positions after the lysine of the functional dyad [71]. The activity toward Kv1.2 is attributed to a lower overall toxin charge, fewer basic residues in the C-terminal region, Tyr as the aromatic residue of the functional dyad, Ile two positions after the functional Lys, and basic/Gln residues in positions 19/20 [56].

In this sense, Tf5 presents characteristics like those observed in other toxins from the same group, such as the presence of two Lys residues in positions K27 and K33, which, upon contact with the channel, lead to the projection of Lys27 into the pore, thus promoting the interruption of the potassium flow in the selectivity filter [55]. Further, Tf5 possesses a Tyr residue located nine amino acids away from K27, which favors activity on Kv1 channels, and a conserved K-C-X-N motif in the C-terminal half of its sequence, with the position X occupied by methionine. A comparison of the Kv1 channel blocking activity of KTxs differing precisely at residue X has shown that the toxins with Met preferentially blocked the Kv1.1 and 1.6 subtypes [66]. The exceptions are α-KTx4.2 (Ts9) [72] and α-KTx4.8 (Cm39) [55], which have threonine and isoleucine at this position, respectively. Tf5, with Met in position X of the conserved C-terminal motif, has not been tested on Kv1.6, but it blocked Kv1.3 at 500 nM, while not presenting any activity on Kv1.1. Considering the determinants for selectivity between Kv1.2 and Kv1.3, it was expected that the Tf5 toxin would be more selective for Kv1.3, as it exhibits an overall positive charge, basic residues in its C-terminal, and a Met two positions after the critical Lys. However, based on the results obtained, it is suggested that the determinants for Kv1.2, such as the tyrosine of the functional dyad, play a more significant role in this selectivity.

Tf5 shares significant sequence identity with all KTx4 peptides, and around 50% of its identity with scorpion KTxs belonging to other α-KTx families. Notably, as revealed by its sequence alignment, Tf5 exhibited pronounced similarities to α-KTx4.1 (*Tityus serrulatus* Ts7, 94.59%), α-KTx4.5 (from *T. costatus,* 94.59%), α-KTx4.6 (from *T. stigmurus* Tst26, 91.89%), and α-KTx4.7 (from *T. stigmurus*, 94.59%), toxins belonging to Brazilian *Tityus* species whose venom contains Na^+^ channel modulators that have high levels of shared sequence identity as well [73].

The α-KTx4.8 (Cm39) from *Centruroides margaritatus* has a lower number of basic amino acids, a lower overall charge of 3.6, a Thr (T34) instead of Lys, and Ile (I31) instead of Met, all of which confer selectivity for the K_V_1.2 channel [55].

α-KTx4.6 (Tst26) electrophysiology assays reported activity on the Kv1.2 and Kv1.3 channels, observing K_D_ values of 1.9 nM and 10.7 nM, respectively, and an absence of activity on the Kv1.1, Kv1.4, Kv1.5, Kv11.1, hIKCa1, hBK, and hNav1.5 channels at a concentration of 10 nM [56]. α-KTx4.6 (Tst26) and Tf5 differ in three residues: at position 6, Tf5 has an arginine instead of a Lys; at position 19, it has a Glu instead of Gln; and at position 24, it has a Ser in place of Ala. The first substitution, K6R, does not change the charge and should have a minor impact on toxin selectivity. The Glu in the Gln position may be critical for the potency of Tf5 in the Kv1.2 and Kv1.3 channels, as it replaces a non-charged polar residue with a negatively charged acidic residue. The third substitution, which occurs at position 24, replaces an Ala, a hydrophobic residue, with a non-charged polar Ser, three positions before the Lys of the functional dyad (Lys27), possibly hindering the interaction of Lys with the ion pore to block conductance.

Another determinant in the primary structure proposed to affect selectivity to Kv1.2 or Kv1.3 is position 19 [56]. Whether a Gln or basic residue in position 19 leads to a higher affinity for Kv1.2 in α-KTx4 toxins can be questioned by comparing the sequence of α-KTx4.6 with α-KTx4.1 and the Tf5 toxin. α-KTx4.1 differs from α-KTx4.6 only in position 19, where α-KTx4.1 has Glu instead of Gln. Experiments on α-KTx4.1 by Rodrigues et al. (2003) [57] reported a K_D_ of 3.9 nM at pH 7.5 in two-electrode voltage clamp experiments on oocytes expressing mKv1.3. Patch clamp experiments on L929 cells stably transfected with Kv1.3 showed a K_D_ of 19.8 nM using the same methodology applied in this study. For Kv1.2, a K_D_ of 0.21 nM was observed with α-KTx4.1 [74]. Papp et al. proposed the K_D_ ratio of Kv1.2 and Kv1.3 (K_D_Ratio = K_D_Kv1.2/K_D_Kv1.3) to assess selectivity for Kv1.2 in this toxin family. The α-KTx4.6 toxin has a K_D_ ratio of 0.18, while α-KTx4.1 has a K_D_ ratio of 0.01–0.05 [56], indicating that the substitution of Gln with Glu increased selectivity for Kv1.2. The presence of Glu in position 19 in the Tf5 toxin maintained its selectivity for Kv1.2, as observed with the K_D_Ratio of 0.13 of Tf5.

So far, it is known that a great number of peptide toxins derived from scorpions block K^+^ channels with high affinity [75]. Properly assessing the selectivity of a toxin for a specific ion channel requires precise quantitative parameters. As proposed by Giangiacomo (2004) [76], a difference of at least 100-fold in the dissociation constant (Kd) between a target channel and other channels is a possible criterion by which to define selectivity. Based on this criterion, the Tf5 toxin interacts with the Kv1.2 and Kv1.3 channels; however, the lack of a Kd difference greater than 10-fold indicates that Tf5 does not exhibit selectivity towards these channels.

α-KTx4.1, at a concentration of 1 µM, inhibits the Kv1.1, Kv1.2, and Kv1.3 channels with inhibition rates of 85%, 91%, and 94%, respectively [77]. Interestingly, it is active on Kv1.1, whereas the α-KTx4.6 toxin does not exhibit activity on this channel. However, it is important to note that the experiments with α-KTx4.6 on Kv1.1 were conducted at 10 nM, and this difference in concentration could account for the absence of activity. Additionally, Tf5 exhibited 94.59% similarity with the toxin α-KTx4.5, which at 20 uM presents blocking activity on the following channels (in order of specificity): Kv1.3 > Kv1.2 > Kv1.1 > ERG > NR2a (the NMDA receptor subunit (NR2A)) [49].

The therapeutic use of native toxins in the treatment of channelopathies may be unfeasible when they act on more than one ion channel or molecular target. However, structure–function studies can provide relevant information for the design of more selective analogs. The chemical and electrophysiological characterization of K^+^-channel blockers isolated from scorpion venoms has been and will continue to be of great importance for understanding the functioning of ion channels, as well as a promising source of new drugs. One of the primary reasons why peptides with therapeutic potential fail to advance to clinical trials is their toxicity, particularly their hemolytic effects [78]. In silico analyses predicted that alpha-KTx4.9 would not exhibit cytotoxic activity, a characteristic previously described in KTx only within β-KTx family scorpions, likely due to its N-terminal motif [79].

Some Kv1-channel-blocking toxins have already demonstrated therapeutic properties for autoimmune diseases, chronic inflammatory conditions, and certain types of cancer. MgTx, isolated from the scorpion *Centruroides margaritatus*, is a Kv1.2 and Kv1.3 blocker [65] that has been shown to reduce the viability of A549 carcinoma cell cultures [80]. ShK-186, commercially known as Dalazatide, is a selective Kv1.3 blocker that has undergone phase 1 clinical trials for the treatment of plaque psoriasis [81].

## 4. Conclusions

In this work, a new purified α-toxin from *Tityus fasciolatus* scorpion venom has been chemically and electrophysiologically characterized using different voltage-dependent potassium channels. It has been shown that Tf5 can block the voltage-dependent potassium channels Kv1.2 and Kv.1.3 with an IC_50_ of 15.53 ± 1.12 nM and 116.41 ± 1.15 nM, respectively, while not displaying a significant blockade of the Kv1.1 and Kv1.4 channels, thus contributing to our understanding of the requirements for scorpion α-KTx and K^+^ channel interactions.

## 5. Materials and Methods

### 5.1. Venom Extraction

Scorpions were collected in Brasilia, Brazil, under IBAMA license number 048/2007—CGFAU. *Tityus fasciolatus* specimens were kept at the University of Brasilia, with water provided ad libitum, and fed monthly. Venom extraction was performed by electrical stimulation (12 V). The venom was collected in ultrapure water and centrifuged at 10,000× *g* for 15 min. The supernatant was collected, quantified by its absorbance at 280 nm and 260 nm, and dried using a SpeedVac (ThermoFischer, Waltham, MA, USA). To quantify the amount of protein in the crude venom, we used the following equation from the Warburg–Christian method [82]:C(mg·mL^−1^) = (1.55 (Abs280)) − (0.76(Abs260)) (1)

### 5.2. Construction of the cDNA Library and Gene Cloning

The *T. fasciolatus* cDNA library was constructed from total RNA obtained from venom glands that were dissected from two animals (one male and one female) 4 days after venom extraction by electrical stimulation. The total RNA was extracted using the ZR-Duet DNA/RNA Miniprep kit (Zymo Research, Irvine, CA, USA), and from this the full-length cDNA library was prepared using the In-Fusion SMARTer cDNA Library Construction kit (Clontech Laboratories, Palo Alto, CA, USA) according to the long-distance polymerase chain reaction (PCR) protocol. The cDNA inserts were cloned into linearized pSMART2IF plasmids, and the recombinant plasmids were transformed into electrocompetent *Escherichia coli* DH5α cells. Then, 10 μM of both the forward and reverse screening primers (Advantage 2 PCR kit, Clontech Laboratories, Palo Alto, CA, USA) was used for the PCR screening of the cDNA library. Selected plasmids with >300 bp cDNAs were isolated using the alkaline lysis method, and single-pass sequencing of their 5′ termini was conducted with the M13 forward primer (5′-TGT AAA ACG ACG GCC AGT-3′) on an automatic sequencer (ABI 3130 XL genetic analyzer, Applied Biosystems, Foster City, CA, USA) according to the manufacturer’s instructions. Bioinformatics analyses were performed as previously described [73]. The nucleotide sequence data reported in this paper were deposited in The National Center for Biotechnology Information (NCBI) GenBank as entry PQ540980.

### 5.3. Toxin Purification

Crude *T. fasciolatus* venom was fractionated by RP-HPLC, Reverse-Phase High-Performance Liquid Chromatography (Shimadzu LC 20-AD), using a linear gradient that moved from 100% of solution A (0.12% trifluoroacetic acid—TFA—in water) to 60% of solution B (0.10% TFA in acetonitrile) at a flow rate of 1 mL/min for 60 min, using a C18 Semi-Preparative column (Jupiter 5 µm C18 300 Å, 250 × 10 mm, Phenomenex, Inc., Torrance, CA, USA) and monitoring the absorbances at 216 nm and 280 nm. The fraction that eluted at 33.5% acetonitrile was purified by RP-HPLC using an analytical C18 column (250 × 4.60 mm, 4 µm, Phenomenex, Inc., Torrance, CA, USA), within a linear gradient from 30% to 40% of solution B at a flow rate of 1 mL/min for in 20 min. Tf5 was recovered at 33% acetonitrile.

### 5.4. Molecular Mass Analysis and Amino Acid Sequencing

The molecular mass, purity, and partial sequence of the toxin were determined by matrix-assisted laser desorption ionization time-of-flight (MALDI TOF/TOF) spectrometry in an Autoflex Speed (Bruker Daltonics, Billerica, MA, USA, EUA) and using FlexControl (version 3.4) and FlexAnalysis (version 3.4 Build 76, Bruker Daltonics) software in their positive linear mode to obtain its average molecular mass and positive reflector mode to obtain its monoisotopic mass.

For mass determination, samples were mixed with α-cyano-4-hydroxycinnamic acid (HCCA) (10 mg/mL) and plated on an Anchorchip stainless steel sample plate (Bruker Daltonics, Germany), and then allowed to dry at room temperature. For partial sequence determination, the sample was diluted in a 1,5-diaminonaphthalene solution (1:1; *v*:*v*) and analyzed by the In-source Decay (ISD) method.

### 5.5. Bioinformatics Analysis

The experimental molecular mass was compared to the theoretical mass of the putative mature toxin deduced from the precursor sequence translated from the *Tityus fasciolatus* venom gland’s cDNA library. The signal peptide of the precursor sequence was predicted on the SignalP 4.1 Server (http://www.cbs.dtu.dk/services/SignalP/, accessed on 5 January 2025).

For the structural prediction of the peptide, the AlphaFold machine learning model was used, which is an artificial intelligence-based model used for protein structure prediction. The sequence was submitted, and several folding models were obtained, from which the one with the highest reliability scores was selected, with a pTM = 0.72 and a pLDDT score (the per-atom confidence estimate) higher than 90 [83].

A hemolytic in silico assay was performed using the ToxGIN machine learning algorithm [53], dispensed as an online tool at https://dbaasp.org (accessed on 11 February 2025), and the HLPpred-Fuse algorithm [54].

### 5.6. Alignment

The predicted mature peptide sequence was submitted to the ExPASy server (https://web.expasy.org/compute_pi/, accessed on 5 January 2025) for theoretical molecular mass calculation and to BLASTP (https://blast.ncbi.nlm.nih.gov/Blast.cgi, accessed on 20 December 2024) for a sequence similarity search. For sequence alignment, Clustal Omega (https://www.ebi.ac.uk/Tools/msa/clustalo/, accessed on 20 December 2024) was used.

### 5.7. Electrophysiological Assays

#### 5.7.1. Cell Culture

Human Embrionary kidney HEK293T cells were grown in Dulbeccos’s Modified Eagle’s Medium (Sigma-Aldrich, St. Louis, MO, USA) supplemented with 10% fetal calf serum (GIBCO-ThermoFischer, Waltham, MA, USA) and 1% Antibiotic-Antimycotic^®^ (GIBCO-ThermoFischer, Waltham, MA, USA). HEK293T cells were transfected with 1 μg of pcDNA3-Kv1.1, pcDNA3-Kv1.2, or pcDNA3-Kv1.4 and 0.1 μg of EGFP (Clontech Laboratories, Palo Alto, CA, USA) using Lipofectamine 3000 (ThermoFischer, Waltham, MA, USA). L929 cells expressing the Kv1.3 channel were grown in Dulbeccos’s Modified Eagle’s Medium (Sigma-Aldrich) supplemented with 10% fetal calf serum and 100 units/mL of penicillin and 10 µg/mL of streptomycin. Cells were grown in an incubator with a moist atmosphere and 5% CO_2_, at a constant temperature of 37 °C.

#### 5.7.2. Current Recording

Potassium macrocurrents (IKs) were measured in whole-cell mode using a HEKA EPC 10 USB amplifier and controlled by PATCHMASTER software version 2x92 (HEKA Elektronik, Lambrecht, Germany). Experiments were performed at a room temperature of 23 °C. The pipettes were made from Borosilicate glass capillaries, with a 100 mm diameter, OD. Of 1.5 mm, ID. of 0.84 mm, and pulled in a horizontal puller P97 (Sutter Instruments, Novato, CA, USA). After filling them with the pipette solution and placing them inside the bath solution, their resistance was between 1.8 and 2.5 MΩ. Their series resistance was approximately 10 MΩ, which was compensated at least 60%. The p/4 protocol, with a hold potential of −100 mV, was applied to cancel out the capacitive and leak currents. Currents were stimulated at 30 mV over 200 ms, from a holding potential of −80 mV, in intervals of 20 s.

#### 5.7.3. Solutions and Toxin Preparation

The pipette (internal) solution contained (mM) KCl 140, MgCl_2_ 2, CaCl_2_ 1, EGTA 10, and HEPES 10, with a pH of 7.3 corrected with KOH and an experimental osmolarity of 337 mOsmol. The bath (external) solution contained (mM) NaCl 155, KCl 4.5, MgCl_2_ 1, CaCl_2_ 2, and HEPES 5, with a pH of 7.4 corrected with NaOH and an experimental osmolarity of 307 mOsmol. The solutions’ osmolarity was measured with a K-7400S semi-micro osmometer (Knauer Berlin, Germany). Tf5 was diluted using the external solution and added into the extracellular bath to achieve the desired concentration.

## Figures and Tables

**Figure 1 toxins-17-00096-f001:**
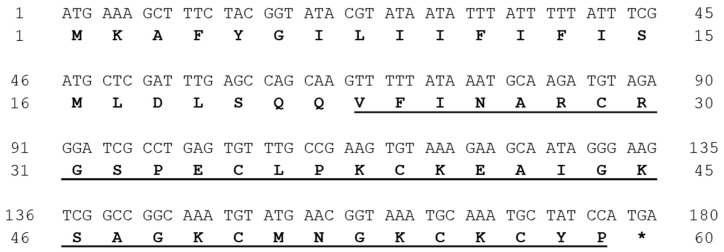
Cloned nucleotide and deduced precursor amino acid sequences for Tf5. The predicted signal peptide (SP) is highlighted in bold and the mature peptide is in bold and underlined. The right column shows the consecutive numbering of the sequences, starting at the ATG for the cDNA and the first amino acid of the SP for the protein. Indicated with an asterisk (*) is stop codon.

**Figure 2 toxins-17-00096-f002:**
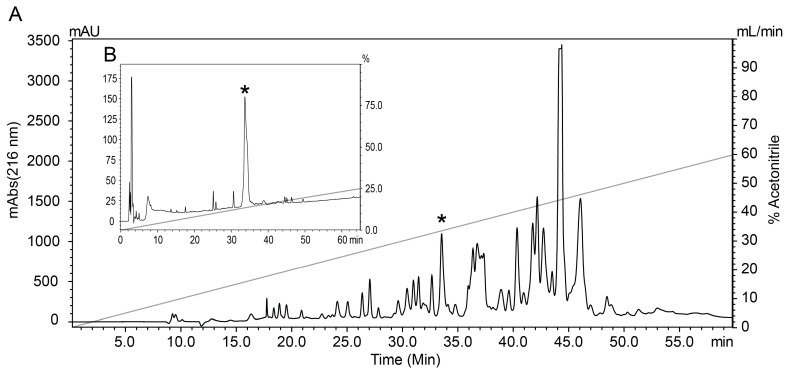
Purification of Tf5. (**A**) Chromatogram showing the separation of the crude *T. fasciolatus* scorpion venom (3.5 mg) over a linear gradient of 0 to 60% acetonitrile, in 60 min. Indicated with an asterisk (*) is the fraction containing the Tf5 peptide. (**B**) The second purification step, from which the pure Tf5 peptide was obtained, eluting at 33% acetonitrile.

**Figure 3 toxins-17-00096-f003:**
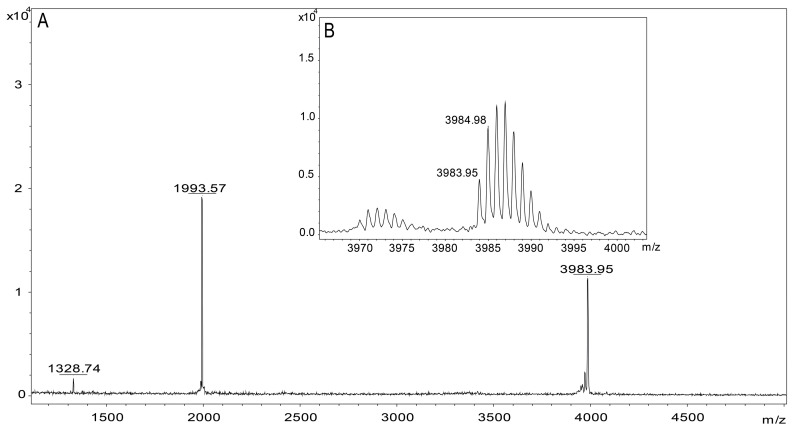
Mass spectrometry analysis of Tf5 by MALDI-TOF. (**A**) Monoisotopic masses of Tf5, [M + 2H]^2+^ = 1993.57 Da, [M + 3H]^3+^ = 1328.74 Da, and [M + H]^+^ = 3983.95 Da. (**B**) Isotopic mass pattern of Tf5, [M + H]^+^.

**Figure 4 toxins-17-00096-f004:**
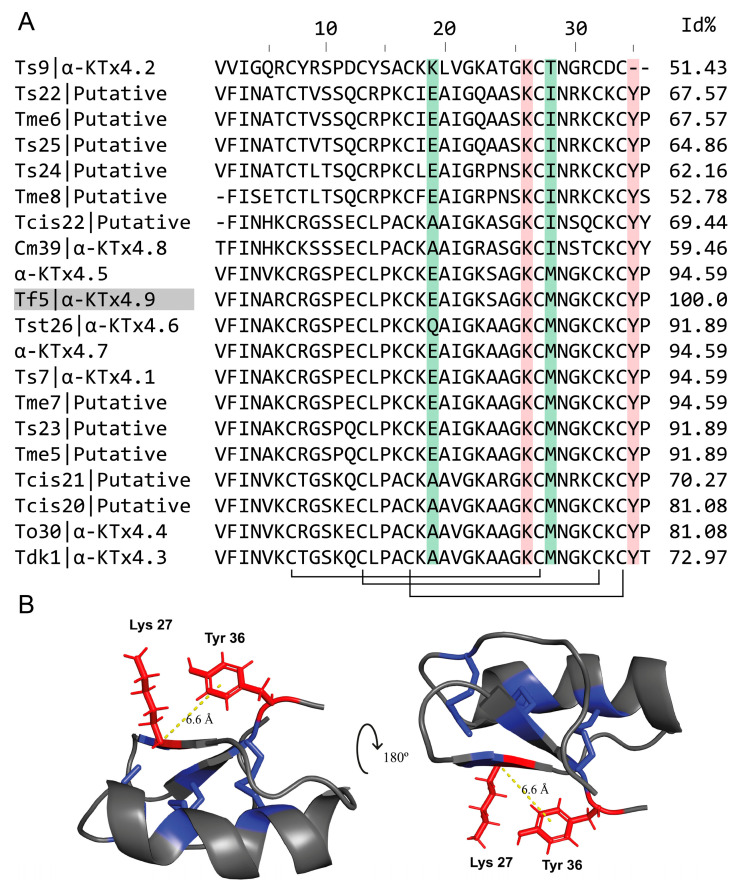
Multiple Sequence Alignment and a model of Tf5. (**A**) Sequence alignment of Tf5 with other Tityus peptides: α-KTX4.9 (Tf5 from *T.fasciolatus*), α-KTX4.7 (from *T. stigmurus*), α-KTX4.5 (from *T. costatus*), α-KTX4.1 (Ts7 from *T. serrulatus*), α-KTX4.6 (Tst26 from *T. stigmurus*), α-KTX4.4 (To30 from *T. obscurus*), α-KTX4.3 (Tdk1 from *T. discrepans*), α-KTX4.8 (Cm39 from *Centruroides margaritatus*), α-KTX4.2 (Ts9 from *T. serrulatus*), and putative sequences (Ts22, Ts23, Ts24, and Ts25 from *T. serrulatus*; Tme6, Tme7, and Tm8 from T. melici; Tcis20, Tcis21, and Tcis22 from *T. cisandinus*). Red highlights the functional dyad and green highlights some possible determinants of selectivity for Kv1.2/Kv1.3. (**B**) The three-dimensional structure of Tf5 consists of three β-strands (Val3-Lys8, Gly28-Met31 and Lys34-Cys37), aligned to form an antiparallel β-sheet, that are anchored to a single α-helix (Cys15-Gly24) by three disulfide bonds (marked in blue).

**Figure 5 toxins-17-00096-f005:**
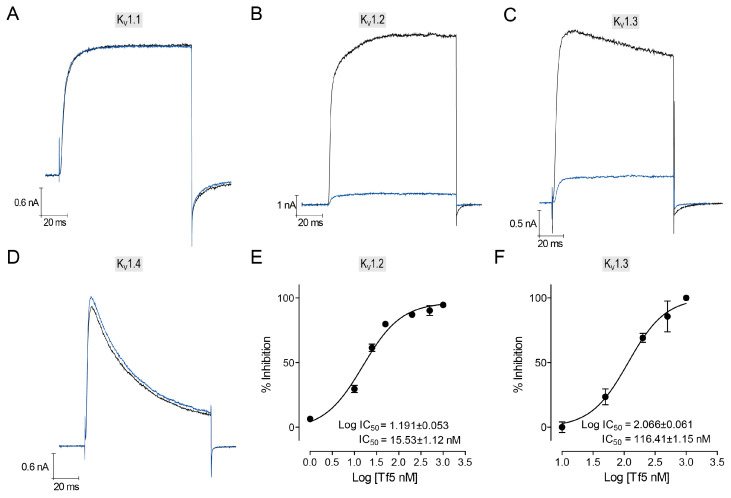
Electrophysiological characterization of Tf5 toxin using potassium isoforms. The Tf5 toxin was tested at 500 nM on four different voltage-dependent potassium channel subtypes: Kv1.1 (**A**), Kv1.2 (**B**), Kv1.3 (**C**), and Kv1.4 (**D**). Among these, toxin activity was observed in the Kv1.2 and Kv1.3 channels, which demonstrated 90.23 ± 3.78% and 81.80 ± 4.29% current inhibition, respectively. Panels (**E**,**F**) show the dose–response curves for Tf5 obtained on Kv1.2 and Kv1.3, respectively, yielding IC_50_ of 15.53 ± 1.12 nM and 116.41 ± 1.15 nM. Black traces show the control condition and blue traces the Tf5 500 nM condition. The number of replicates (n) for each concentration was ≥5.

## Data Availability

The original contributions presented in this study are included in this article and its Appendix A. Further inquiries can be directed to the corresponding author.

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
