# Peer review of "The First K+-Channel Blocker Described from Tityus fasciolatus Venom: The Purification, Molecular Cloning, and Functional Characterization of α-KTx4.9 (Tf5)"

_toxins, 2025, doi:10.3390/toxins17020096_

Round 1
Reviewer 1 Report
Comments and Suggestions for Authors
This manuscript describes the identification, purification, and characterization of Tf5, the first potassium channel-blocking peptide isolated from the venom of Tityus fasciolatus, an endemic Brazilian scorpion. Through a combination of transcriptomic analysis and protein purification, the authors identified a novel 37-amino acid peptide with a molecular mass of 3,983.95 Da. The peptide's structure was investigated using both homology modeling based on α-KTx4.5 and AlphaFold predictions, showing the characteristic cysteine-stabilized α/β scaffold common to scorpion K+ channel toxins. Functional studies demonstrated that Tf5 selectively blocks Kv1.2 and Kv1.3 channels with IC50 values of 15.53 nM and 116.41 nM respectively, while showing no significant activity against Kv1.1 and Kv1.4 channels. Based on sequence homology and structural characteristics, Tf5 was classified as α-KTx4.9, representing a new member of the α-KTx4 subfamily.
However, there are some inconsistencies in the methodology description that need to be addressed. The authors state in the results section that "The Tf5 tertiary structure was modeled based on the x-ray-determined structure of alpha-KTx4.5 (pdb 6atn.1), which shares 94.59% identity with Tf5," while in the methods section they indicate that "For the structural prediction of the peptide, the AlphaFold machine learning model was used." While both approaches are valid and potentially complementary, this methodological ambiguity should be clarified to better understand the structural analysis workflow.
This observation leads to a significant opportunity for enhancing the manuscript's impact. Using the AlphaFold server, this reviewer was able to obtain docked poses of the Tf5 toxin onto the Kv1.2 and Kv1.3 channels, while Kv1.4 failed to produce bound models, and Kv1.1 displayed some distortions. These computational experiments, completed within 10 minutes, suggest that relatively modest additional work could provide substantial structural insights. Given the authors' demonstrated proficiency with AlphaFold, I strongly recommend adding a figure showing the docked complexes (Kv1.2, Kv1.3) and comparing them to the experimental complex structures of CTX (PDB: 4JTA), DTX (PDB: 8VC3) and Shk (7SSV). Such analysis would elevate the discussion beyond sequence-based selectivity determinants to provide mechanistic insights at the structural level.
Minor issues:
While the dose-response curves for Kv1.2 and Kv1.3 inhibition are well presented in Figure 5, the IC50 values (15.53 nM and 116.41 nM respectively) are reported without standard deviations, despite clear error bars being present in the figures. These statistical parameters as well as the number of replicates should be indicated in the figure legend.
In Line 241 – “Iso two positions after the functional Lys” – should probably read “Ile two positions after the functional Lys”.
Reviewer 2 Report
Comments and Suggestions for Authors
This study identifies Tf5, a novel peptide from the venom of the Brazilian scorpion Tityus fasciolatus, which significantly blocks Kv1.2 and Kv1.3 potassium channels. Purified Tf5, with a molecular mass of 3,983.95 Da, shares a high degree of sequence identity with the α-KTx4 subfamily and is classified as α-KTx4.9. This research is noteworthy as it introduces the first Kv channel blocker from T. fasciolatus, underscoring its potential therapeutic applications. The methodology is well-designed and effectively addresses the primary scientific question, with conclusions firmly rooted in the findings. Given the biochemical nature of Tf5, it is worth investigating its potential haemolytic properties. The authors might consider using computational tools to explore this aspect, thereby enhancing our understanding of the therapeutic potential of this bioactive compound from scorpion venom.
1. The abstract section needs an introductory sentence and aims to establish the state of art and the research objectives.
2. lines 10-11. The word conferring is not appropriate in this context.
3. The perspectives and importance of the research were not highlighted in the abstract section.
4. Please read the instructions for authors on the journal’s website. The authors must include key contributions section.
5. Some sentences must be supported by references. For example, lins 22-23, 32-33, 36-37, 79-81.
6. The last paragraph of the introduction repeats results that are already discussed in the abstract and results section. I suggest removing this repetition and instead ending the introduction by clearly stating the aims and significance of the study.
7. Lines 105-108. Please avoid a one-sentence paragraph. Expand the idea or connect paragraphs with similar context.
8. Lines 112-113. What is the purity level of the peptides? The chromatogram displays multiple peaks (figure B), suggesting impurity. Have the authors conducted a third chromatographic run to confirm the purity levels? The manuscript does not mention the purity level, which is a crucial parameter for biological assessment.
9. Lines 144-118. Please avoid a one-sentence paragraph. Expand the idea or connect paragraphs with similar context.
10. Please avoid repeating the figures in the discussion section.
11. Figure 5D was not mentioned in the manuscript.
12. Have the authors assessed the toxicity of these peptides? If insufficient material is available, I recommend using computational tools, which are straightforward and freely accessible. This paper with DOI: 10.3390/ph15030323 can help identify appropriate tools and also expand the discussion, particularly when the authors discussed the therapeutic potential use (lines 312-316). Assessing the selectivity is relevant in the context of the development of peptide-based therapeutics. The peptide is rich in basic amino acids, which can disrupt red blood cell membranes. If this is confirmed, novel studies for refinement and reducing toxicity can be suggested for example.
13. Lines 267-269. Please avoid a one-sentence paragraph. Expand the idea or connect paragraphs with similar context.
14. Line 339. What is the reference for this equation? Did the authors develop it themselves, or is it based on previous studies?
15. Lines 369-373. Please avoid a one-sentence paragraph. Expand the idea or connect paragraphs with similar context.
16. Please compare the IC50 with the literature.
Round 2
Reviewer 2 Report
Comments and Suggestions for Authors
The authors have successfully addressed all major points raised by this reviewer. However, a primary limitation remains the absence of quantification for the purity level, even though mass spectrometry results indicate high purity. Despite this, the data is compelling and significantly contributes to the field.
The authors have predicted the haemolytic properties of the peptide, yet this information is absent from the manuscript. Including this finding in the results section would be beneficial. Furthermore, the discussion should explore the selectivity and potential toxicity, as these aspects are crucial. Additionally, the methodology employed for this prediction should be detailed in the methods section. Highlighting this approach will encourage authors to integrate in silico methods in their experimental designs. It is important to note that, occasionally, predictions may contradict experimental findings. Such discrepancies are valuable for refining algorithms and developing novel models. Overall, I strongly support the publication of this study, provided these minor recommendation are addressed.
